Assessing local impacts of the A.D. 1700 Cascadia earthquake and tsunami using tree ring growth
histories: A case study in South Beach, Oregon, U.S.A.
Robert P. Dziak[1], Bryan A. Black[2], Yong Wei[3], and Susan G. Merle[4]
[1]NOAA/Pacific Marine Environmental Laboratory, Newport, Oregon, 97365 U.S.A.
[2]Laboratory of Tree-Ring Research, University of Arizona, Tucson, Arizona, U.S.A.
[3]NOAA/Pacific Marine Environmental Laboratory, Seattle, Washington, 98115 U.S.A.
[4]Cooperative Institute for Marine Resources Studies, Oregon State University, Newport, Oregon, 97366 U.S.A.
*Correspondence to*: Robert P. Dziak (Robert.p.dziak@noaa.gov)
**Abstract.** We present an investigation of the disturbance history of an old-growth Douglas-fir (*Pseudotsuga menziesii*) stand
in South Beach, Oregon for possible growth changes due to tsunami inundation caused by the A.D. 1700 Cascadia Subduction
Zone (CSZ) earthquake. A high-resolution model of the 1700 tsunami run-up heights at South Beach, assuming an "L" sized
earthquake, is also presented to better estimate the inundation levels several kilometers inland at the old-growth site.  This
tsunami model indicates the South Beach fir stand would have been subjected to local inundation depths from 0 to 10 m.
Growth chronologies collected from the Douglas-fir stand shows that trees experienced a significant growth reductions in the
year 1700 relative to nearby Douglas-fir stands, consistent with the tsunami inundation estimates.  The +/- 1-3 year timing of
the South Beach disturbances are also consistent with disturbances previously observed at a Washington state coastal forest
~220 km to the north. Moreover, the 1700 South Beach growth reductions were not the largest over the >321 year tree
chronology at this location, with other disturbances likely caused by climate drivers (e.g. drought or windstorms). Our study
represents a first step in using tree growth history to ground-truth tsunami inundation models by providing site specific physical
evidence.

## 1.  Introduction

Recent studies have demonstrated the utility of using tree-ring growth chronologies for assessment of tsunami and
earthquake impacts on coastal environments [Buchwal and Szczucinski, 2015; Kubota et al., 2017; Wang et al., 2019].
Catastrophic tsunami inundation events along the Sumatra and Japan coasts have shown tsunamis can have a
devastating effect on coastal forests and overall coastal geomorphology [Kathiresan and Rajendran, 2005; Udo et al.,
2012; Lopez Caceres et al., 2018]. In addition to the physical impacts from tsunamis, Kubota et al. 2017 showed that
coastal trees that survived direct physical damage from the great 2011 Japan began to die the following summer,
likely due to the physiological stress of salt water immersion.  Wang et al (2019) performed a regional assessment of

coastal western Washington forests and demonstrated that seawater exposure drives reductions in growth, increased mortality and greater climate sensitivity, regardless of whether the seawater exposure is recent or long-term.

Ground motion caused by the megathrust earthquake can also cause significant forest disturbance by toppling trees, damaging root systems, severing limbs and crowns, inducing damaging landslides, or altering the hydrology of a stand, among other potential effects [e.g. Shepard and Jacoby, 1989]. These disturbances appear in the tree-ring record of surviving trees as sudden growth suppression events (when there is damage), or growth increases in the case of reduced competition from adjacent damaged trees.

Here we present an investigation of the disturbance history of an old-growth forest in South Beach, Oregon (**Figure 1**). We also present a new, high resolution model of the 1700 tsunami run-up heights at South Beach to better estimate the inundation levels at the site of the old-growth forest. Our goal is to use tree-growth to ground-truth the tsunami impacts and inundation levels as well as for insights into the degree of shaking caused by the 1700 magnitude 9.0 Cascadia Subduction Zone (CSZ) earthquake [Satake et al., 2003; Witter et al.., 2011].

Interestingly, direct evidence of seismic shaking (liquefaction, landslides, etc) from the 1700 CSZ earthquake is relatively rare along the Oregon Coastal Range [Struble et al., 2020]. This is thought to be due to the high rainfall and water erosion rates in the Pacific Northwest which removes liquefaction evidence in coastal estuaries, and makes landslides in the coast range difficult to identify [Yeats, 2004; LaHusen et al., 2020]. Models of shaking and ground motion along the Oregon coast during the 1700 CSZ earthquake indicate it should have been violent and widespread [WDNR, 2012], and it is plausible that evidence of this shaking might be recorded in the in the form of traumatic resin ducts and ring width suppression of trees along the coast. Very little tree-ring work has been conducted along the Oregon coast; the vast majority of tree-ring research in the Pacific Northwest has entailed climate reconstructions from high-elevation sites in the Cascade Mountains and Olympic Peninsula where competitive effects are relatively lower. We sampled a mesic old-growth forest near the Pacific coast where competitive effects are high. Significant disturbances from the 1700 earthquake and tsunami should substantially alter radial growth patterns as some trees are damaged or killed and resources are redistributed to survivors. Alternatively, the tsunami may cause physical damage to trees resulting in growth reductions. The goal of this study is to investigate whether these disturbances are observable in the few remaining old-growth forests along the coast of Oregon. Thus, we chose a site where good inundation models exist, and there is significant public concern about tsunami impacts because of the presence of a large population (> 10,000 people) and municipal infrastructure.

**2.0 Evidence for Megathrust Earthquakes and Tsunamis**

On 26 January at 9:00PM,1700 A.D., a large earthquake occurred along the Cascadia Subduction Zone, the interface between the Pacific and North American plates along the coasts of California, Oregon, Washington, and British Columbia [Satake et al., 2003]. The earthquake created a tsunami with 10-12 m run-up heights that struck the Pacific Northwest and propagated across the Pacific to Japan [Atwater, 1992; Satake, et al, 2003; Goldfinger et al., 2003]. It is estimated to have most likely been a moment magnitude (Mw) 9.0, with between 13-21 m of coseismic slip on an offshore fault 1100 km long [Satake et al., 2003; Witter et al., 2011]. The 1700 earthquake was preceded by an earthquake in ~960 A.D. (740 yr interval) and another in ~750 A.D. (210 yr interval), with three additional subduction events before these that makes up a recent cluster of 6 megathrust events over the past 1500 yrs [Atwater et al., 2003]. During the 1700 Cascadia earthquake, ground motion and peak ground acceleration (PGA), are modeled from ~0.5-1.2 g along the Oregon coast [WDNR, 2012]. Thus the shaking during this event would have been violent and widespread.

As a result of subsidence, some coastal forests dropped below sea level and were flooded. Boles and root masses of these trees still remain and can be found from northern Oregon to southern Washington. Radiocarbon dating of this wood showed the earthquake occurred around 1700. However, aligning growth patterns of adjacent living trees with those of the flooded, dead trees consistently showed that the last year of growth was 1699, indicating the earthquake occurred between October 1699 and April 1700 [Yamaguchi et al., 1997]. The exact origin time of the earthquake was estimated by calculating the travel time for an unexplained tsunami that struck Japan on 26 January 1700 [Satake et al., 2003; Atwater, 2006]. Surviving trees also recorded the earthquake's date by anomalous changes in ring width or wood anatomy [Atwater and Yamaguchi, 1991; Jacoby et al., 1997]. This tree-ring and dating evidence for coastal disturbance is indeed compelling, however the evidence was derived from trees along just 100 km of coastal southern Washington-northern Oregon, or ~5% of the coastline expected to be affected by a Cascadia megathrust earthquake.

Additional methodologies have been employed to assess the coast-wide impacts of the 1700 earthquake. For example, a coastal-wide inventory of liquefaction features associated with the 1700 earthquake found no features along the Oregon coast, despite numerous exposures of clean sand deposits that must be susceptible to liquefaction, even at low levels of seismic shaking [Obermeier and Dickenson, 2000]. The locations for these field studies in Oregon were also sites where evidence for great Holocene subduction earthquakes (in the form of crustal subsidence) have been identified [Nelson et al., 1995]. The only liquefaction features identified to date (and thus direct evidence of seismic shaking) were found along the Columbia River 35-50 km east of the coast, and these indicate moderate shaking intensity of 0.2-0.35 g [Obermeier and Dickenson, 2000].

**3.0 Model of A.D. 1700 Tsunami**

| 97  | As a first step in estimating tree disturbance in South Beach, we produced a model of tsunami inundation level and |
| 98  | expected flow speed for the 1700 earthquake based on estimates of size, location, displacement and coastal subsidence |
| 99  | [**Figures 2a,b,c**; Witter et al, 2011]. Thus, the modeled run-up height of the 1700 tsunami can be used as a basis to |
| 100 | investigate possible impacts along the coast and estuaries of South Beach. **Figures 2a,b** show the model results of |
| 101 | tsunami inundation level and flow speed for South Beach assuming the "L" or large sized earthquake (Mw 9.0) for |
| 102 | the A.D. 1700 event [Wei, 2017]. The L model assumes a finite-fault source with maximum vertical coseismic |
| 103 | displacement of 15.2 m and subsidence of ~1.03 m at South Beach [**Figure 2c**; Witter et al., 2011]. The Witter et al. |
| 104 | (2011) coseismic subsidence estimate differs slightly from the Satake et al (2003) estimate of nearly 1 m at South |
| 105 | Beach because it is based on coseismic slip from turbidite records [Goldfinger, 2011], and includes a rupture model |
| 106 | with slip partition into a splay fault in the accretionary wedge. The earthquake source duration was not taken into |
| 107 | account in the model. |

| 108 | |

| 109 | Two models were used to compute the tsunami inundation levels and flow speed [Wei, 2017] based on four-level |
| 110 | one-way nested model grids at the spatial resolutions of 1 arc min (~ 1.8 km), 12 arc sec (~ 360 m), 2 arc sec (~ 60 |
| 111 | m), and 1/6 arc sec (~ 5 m). The tsunami simulation model MOST (Method of Splitting Tsunami) model [Titov and |
| 112 | Gonzalez, 1997] used in this study is based on the shallow-water wave equations and uses the estimates of coseismic |
| 113 | slip to account for deep-water wave generation and propagation. The present MOST code version utilizes the Graphics |
| 114 | Processing Unit (GPU) technology that has led to significant reduction of computational time. The MOST model then |
| 115 | provides the boundary conditions computed from the level-1 grid (**Figure 2a**) for a Boussinesq model [Zhou et al., |
| 116 | 2011], which takes into account wave dispersion when computing the nearshore wave-propagation field and onshore |
| 117 | tsunami inundation in level-2, 3, and 4 grids (**Figure 2b** shows the coverage of level-4 grid). The digital-elevation |
| 118 | model (DEM) and bathymetric grid of Newport-South Beach in level 4 were used in the tsunami inundation models. |
| 119 | The elevation grid is derived from the Digital Elevation Model provided by the Oregon Department of Geology and |
| 120 | Mineral Industries (DOGAMI). This dataset contains lidar data based on DOGAMI Lidar Data Quadrangles for |
| 121 | Toledo South, Newport North, and Newport South. The horizontal datum of the DEM is WGS 84. The vertical datum |
| 122 | is NAVD 1988, and it is then converted to Mean Higher High Water (MHHW) level, which is the vertical datum in |
| 123 | our tsunami inundation models. MHHW is 2.317 m above the NAVD 1988, and 1.185 m above the actual Mean Sea |
| 124 | Level (MSL) at Newport according to the datum information at the National Ocean Service (NOS) tide gauge at South |
| 125 | Beach. Typically, when performing hazard assessments, Mean High Water (MHW) or MHHW is assumed over the |
| 126 | entire duration of tsunami [Wei, 2017], and using MHHW as the vertical datum usually gives a more conservative |
| 127 | estimate of the tsunami impact. In the present study, we prefer to use MHHW, instead of the actual tidal level, as our |
| 128 | model reference level due to: 1) the uncertainty of the time of the event, which is based on estimates from Japanese |
| 129 | records (Satake et al. 1996), and could vary over a window of 1-2 hours; 2) the uncertainty of the earthquake/tsunami |
| 130 | source; and 3) the uncertainty in the amount of sea level change, which is > 0.5 m over the past 300 years based on a |

rate of 1.77 mm annual increase. The impact of these uncertainties on the model could overshadow the difference
between MHHW and the actual tidal level, and adds an additional level of uncertainty to the model results. A
Manning's coefficient of friction of 0.03 is uniformly applied for both the land and ocean components of the tsunami
propagation model. It is an average Manning's coefficient that Chow (1959) proposed for coastal and riverine areas
(0.025-0.033), and for land surface (0.03-0.04). It's worth nothing that this Manning's coefficient has been widely
used in MOST-based tsunami model forecast methodology and hazard assessments (Tang et al. 2009; Wei et al, 2007
and 2013; Titov et al., 2016; Zhou et al., 2011). To more realistically estimate the tsunami impact produced by the
1700 event, we removed the two jetties at the entrance of Yaquina Bay from the model DEMs, which leads to greater
tsunami inundation levels and impact at South Beach. The tsunami model results discussed hereafter are based on the
revised DEMs without the jetties.

The tsunami inundation model presented here indicates the "L" earthquake, with the co-seismic subsidence taken into
account, would produce a tsunami that could inundate South Beach to runup heights up to 17 m (**Figure 2a**), and
inundation depths up to 16 m (**Figure 3a,b**). The height of the water level at the western section of Mike Miller Park
is generally between 12-15 m, and reduces to between 9-12 m on the eastern side. It is important to note that "tsunami
water level" is a term used to describe the elevation reached by seawater measured relative to a stated datum (MHHW
herein). In contrast, "inundation depth" refers to the local water depth, or height of the tsunami above the ground after
taking into account the co-seismic subsidence at a specific location, as shown in Figure 2c. However, there is
significant variation in the topography of South Beach, and several areas are predicted to experience a range of
inundation depths much less than the 16 m maximum.  For example, the model shows the amount of inundation
decreases eastward of the beach, and the location of the old-growth Douglas-fir stand at Mike Miller State Park in
South Beach may be subjected to a range of inundation depth from negligible to as much as 10 m (**Figure 3b**).
Moreover, the South Beach stand would likely have been subjected to flow velocities between 2-10 m s$^{-1}$ (**Figure
2B**). These velocities are lower than most of the westward portions of the South Beach Peninsula because the stand
is located on topography that can be up to 10 m higher elevation than most of the westward terrain. Nevertheless, it
would seem these tsunami current velocities would be high enough to cause significant damage to the South Beach
trees, through the large mass and momentum of this volume of sea water, that would be observable in the tree growth.

Lastly, it is worth noting that the "L" earthquake tsunami model presented here also involves the activation of splay
faults in the overriding plate above the subduction zone.  Motion on these splay faults introduce a larger co-seismic
subsidence along the coastline, and therefore represent a more extreme inundation scenario for the A.D. 1700 event
than previous models. Based on the turbidites records reported by Goldfinger et al. (2011), the "L" and larger
earthquake scenarios occurred four times in the past 10,000 years, and thus is referred as a 2,500-year event, although
the general earthquake size class and associated time interval for an "L" event is estimated to be 800 years by Witter
et al. (2011).

**4.0  Impacts of Earthquakes and Tsunamis Inundation on Tree Growth**

4.1 *Earthquake induced ring growth disturbance*
Although there is evidence for only moderate levels of ground shaking in coastal Oregon and Washington following
the 1700 earthquake [Obermeier and Dickenson, 2000], ground motion during large earthquakes has been shown to
cause significant forest disturbance in other earthquake prone regions. As previously mentioned, these earthquake-
induced disturbances are caused by felling or damaging trees, inducing local landslides, or altering the stand's water
access [Jacoby et al, 1997].  Trees that survive these disturbances can show sudden growth suppression events due to
damage or even sudden growth acceleration events because of reduced competition from nearby damaged trees.
Moreover, pulses in tree recruitment may follow a large earthquake as young trees colonize gaps left by damaged
overstory individuals [Jacoby et al., 1997].

Trees can respond both directly and indirectly to the effects of large earthquakes. Indirect responses can occur due to
coseismic environmental changes. For example, Fuller (1912) noted trees died from flooding during the 1811-1812
New Madrid earthquakes. Wallace and LaMarche (1979) found coast redwoods *(Sequoia sempervirens)* and Douglas-
firs *(Pseudotsuga menziesii)* tilted by the 1906 San Andreas Fault earthquake had reaction wood, formed to right the
tree, starting in 1907. Meisling and Sieh (1980) reported the January 1857 Fort Tejon earthquake caused conifers to
lose their crowns, which reduced ring widths that took many years to return to pre-earthquake growth rates. Jacoby
and Ulan (1983) showed the September 1899 Alaska earthquake caused near shore Sitka spruces *(Picea sitchensis)*
to increase growth because coseismic uplift resulted in less exposure to wind, salt spray, and root-zone erosion.
Finally, in consideration of direct responses to earthquake impacts, Jacoby et al., (1988) analyzed conifer tree-ring
samples near the epicenter of the 1812 San Juan Capistrano earthquake. A total of nine on-fault trees showed drastic
growth reductions in 1813, requiring decades to return to pre-disturbance growth rates.  Similarly, Sheppard and
Jacoby (1989) showed that the 1964 Alaskan earthquake, which caused ~4 m of coseismic uplift, initially induced
growth reduction in Sitka spruces, but the trees eventually responded with wide reaction wood rings in the following
years to regain upright positions. Van Arsdale et al (1998) showed the New Madrid earthquakes of 1811-1812 caused
inundation of bald cypress trees near Reelfoot Lake (Tennessee) which greatly increased radial growth from 1812 to
1819. In contrast, the growth of bald cypress trees in northeastern Arkansas was severely suppressed for almost 50 yr
following the earthquakes. Wells and Yeton (2004) studied the 1929 Buller and 1968 Inangahua earthquakes in New
Zealand, finding clear impacts on tree growth, where swamps on elevated terraces are generally best for preserving
earthquake record because they are not affected by drought or wind.  As for tree growth disturbances due to earthquake

shaking, Fu et al. (2020) showed how the 1950 Zayu-Medog magnitude 8.6 earthquake in the southeastern Tibetan Plateau, influenced tree growth during the period 1950–1955. However, alpine trees were less disturbed than those located at mid and low elevations. Severe growth suppressions occurred during the first three years after the earthquake and were stronger at low elevations.

4.2  *Tsunami induced tree-ring growth disturbance*

Just as trees located near epicenters of large earthquakes can be disturbed and experience growth changes from intense shaking and ground displacement, inundation of a coastal forest by a large tsunami should also have a significant impact. Catastrophic tsunami inundation events along the Sumatra and Japanese coasts showed tsunamis have a devastating effect on coastal forests, damaging trees and severely eroded and alter the beach and estuary geomorphology [e.g. Kathiresan and Rajendran, 2005; Udo et al., 2012; Lopez Caceres et al., 2018]. It is expected that inundation by a tsunami would cause significant ring-growth reduction due to physical impact from the wave, prolonged exposure to salt-water, and from tsunami debris that would also physically impact the tree. There are several studies demonstrating the impact of the inundation of large amounts of seawater and salts on coastal trees after the tsunami (e.g. Kubota et al., 2017; Wang et al., 2019).  These studies showed trees that survived direct physical damage from the tsunami began to die the following summer, likely due to the physiological stress of saltwater immersion. Earlywood that formed in the spring following the tsunami had higher $\delta^{13}C$ values in the rings formed prior to the disaster. In a field survey following the 2010 Chile and 2011 Japan tsunamis, Yoshii et al 2012 that the soil deposits collected in the tsunami-inundated areas are rich in water-soluble ions compared with the samples collected in the non-inundated areas.

In the U.S. Pacific Northwest, when the A.D. 1700 co-seismic tree-ring growth disturbance is considered, it is largely of trees killed by inundation attributed to co-seismic subsidence [e.g. Atwater and Yamaguchi, 1991]. However, Jacoby et al., [1997] were able to find trees that pre-dated the 1700 Cascadia earthquake and survived subsidence and inundation, which is analogous to the tree-ring growth scenario we observed in South Beach, Oregon.

Jacoby et al. [1997] collected cores from 33 living Sitka spruce trees that were established earlier than 1700 (i.e. at least 300 years old) that stand along the western Columbia River between Washington and Oregon (~220 km north of South Beach). While 15 of these trees show some evidence of disturbance at 1700, 5 trees showed no disturbance and the remaining 14 could be in either category. There were both unusual decreases and increases in ring-width in disturbed trees. Disturbed trees also showed water-logging and increasing numbers of traumatic resin canals at 1700 (sap-conducting tubes formed by altered cells), but only two formed reaction wood in response to co-seismic tilting

or flooding. Growth responses occurred over a range of years with clear declines occurring as early as 1698 and as late as 1702 to 1706.

Thus, the exact timing of tree-ring disturbances due to an earthquake and the resulting ground motion, coastal land subsidence and tsunami inundation can vary within a few years around the event date. This is because tree growth can be affected by many climatological/meteorological factors, including droughts, cold/heat stress, fires, and windstorms and even insect infestations. However, comparison of coastal growth rings with other regional sites can be used to control for these climate/weather disturbance impacts. Thus, despite this temporal variability, Jacoby et al (1997) conclude the subduction earthquake/subsidence event occurred between the growing seasons of 1699 and 1700. Therefore it seems likely a combination of the effects from earthquake ground motion, coastal land subsidence, and rapid inundation by several meters of fast-moving sea water can be observed in the variation of ring growth chronologies from trees within the impact zone.

## 5.0  Tree Ring Growth Chronologies from South Beach, Oregon

In an attempt to further quantify the widespread effects of the A.D. 1700 earthquake, we obtained tree-ring records from a stand of old-growth Douglas-fir trees (*Pseudotsuga menziesii*) that pre-date 1700 (**Figure 4a,b**). The stand is located in an Oregon State Park in South Beach, Oregon, roughly 600 m east of Highway 101 (**Figure 3a,b**).  Old growth trees of 300+ years of age are rare along the Oregon coast, thus this stand of trees within the inundation zone presented a unique opportunity to search for direct physical evidence of the impact of a Cascadia Subduction zone earthquake and tsunami inundation in a populated area where tsunami models indicate significant inundation levels and run-up heights.

To ground-truth the model of the A.D. 1700 tsunami, we collected tree cores at breast height from 37 dominant or codominant old-growth trees at the South Beach site using a 32" increment borer. One to two cores were collected from each tree, after which cores were mounted, sanded with increasingly fine lapping film, and cross-dated (Phipps 1985). Each core was then measured using a Velmex TA Tree-Ring Measuring device to the nearest 0.001mm (Velmex, Inc. Bloomfield, NY). Crossdating was then statistically verified using the program COFECHA, which is designed to find errors in chronologies [Holmes 1983]. A master growth-increment chronology was then developed by detrending each measurement time series using a negative exponential or regression functions to retain as much low-frequency variability as possible as well as a second chronology developed using 50-year 50% frequency-cutoff cubic spline to highlight interdecadal to interannual growth variability. All chronology construction was performed using the program ARSTAN, a tree-ring standardization program based on detrending and autoregressive time series modelling [Cook and Krusic 2005].  **Figure 3a,b** shows the location of the Douglas Fir trees sampled for this study

in relation to the modeled tsunami run-up heights for South Beach. Of all the trees sampled, a total of twelves cores
from eight trees pre-dated 1700 (**Figure 4b**).

As noted, the tsunami inundation model presented here (**Figure 2a,b**) indicates the "L" earthquake would produce a
tsunami that could inundate the lowlands of South Beach to inundation depths up to 18 m. However, the Douglas-fir
old-growth stand that is the subject of this study lies on, and along the western edge, of two parallel north-south
striking topographic highs (likely paleo-dune ridge lines). The tsunami model presented here indicates that while
many of the trees in this area may have experienced as much as ~10 m of inundation depth, several trees are also on
high ground and may have experienced much less, or even zero, inundation.

Tree-ring data detrended using negative exponential functions did not reveal major stand-wide releases or
suppressions around 1700 (data not shown) nor did data detrended using the 50-year spline functions. Detailed
examination of the growth-ring samples indicates that although individual cores have below-average growth, and one
experiences what could be interpreted as a post-1700 growth release, variability around 1700 is not necessarily
exceptional in the longer-term context of the ~310 year history of the dataset (**Figure 4a,b**). Indeed, there are several
other growth reductions in the record that are the same magnitude or larger than the disturbance at A.D. 1700. Most
notably, there are large suppressions observed beginning around 1691 and again in 1739 and 1745 (arrows on **Figure**
**4a**). The 1739 reduction has been observed in other old-growth stand chronologies throughout Cascadia and may be
due to a significant climatological event such as a drought [Carroll et al. 2005, 2014].

*5.1 Control Sites in Western Oregon Cascades and Coast Range*

To better detect unusual growth anomalies around 1700, we compared the South Beach Douglas-fir tree-ring data to
two other Douglas-fir data sets from the Oregon Coast Range and one from the western Cascade Mountains, all of
which would have experienced similar climate conditions but not tsunami inundation. The first of these was an old-
growth stand on Marys Peak (~46 km east of South Beach) in the central Coast Range. The second is Browder Creek
located ~160 km east in the western Cascade Mountains [Black et al. 2015; **Figure 5a,b**]. The third was a chronology
generated from dead-sampled trees in lakes in the western and central Oregon Coast Range. These lakes had formed
when landslides impounded streams, and the preserved drowned trees were then used to establish the date of lake
formation, and thus the landslide event [Struble et al. 2020]. Eight lakes had a combined number of 15 trees (and 31
sets of measurements) that pre-dated 1700 and were used to generate a "control" chronology. As with Mike Miller
trees, all measurement time series in the control datasets were detrended with 50-year splines. The Oregon Coast Range
sites range in elevation from 137 m (Hamar Lake) to 380 m (Klickitat Lake), with the exception of Marys Peak, which
is 900 m. The lone western Cascade site, Browder Creek, is a 1108 m. All sites are at low enough elevation that they
are most limited by summer (July – Sept) drought, as opposed to higher elevation sites that are most sensitive to
temperature. Relationships with drought are somewhat stronger at Marys Peak and Browder Creek, but this is likely
due to their more inland locations.

When compared to the control sites, South Beach tree growth is significantly lower than the lakes, Marys Peak, or
Browder Creek in 1700 (t-test of $p < 0.001$) after detrending all series with the 50% frequency cutoff 50-year cubic
splines. Moreover, the lakes trees and Marys Peak trees do not significantly differ in growth in 1700, suggesting the
South Beach Douglas-fir growth is unusually low across sites with the most similar climatic histories and sensitivities.
**6.0 Discussion**

Analysis of tree-ring data presented here indicates there is a reduction in Douglas-fir tree growth at a site associated
with the 1700 Cascadia Subduction Zone earthquake and tsunami in South Beach Oregon relative to other inland sites
in the Oregon Coast Range. The growth reduction is not outside the range of variability, as illustrated by other much
more severe reductions over the 310 year South Beach chronology (**Figure 5a,b**). However, there is at least a subtle
growth reduction that deviates from other nearby locations. Although beyond the scope of this study, further chemical
analysis is needed to show that the tree rings collected at South Beach exhibit higher $\delta^{13}C$ or water-soluble ion levels
to establish the trees here were immersed in seawater.

The tsunami inundation model presented here, which assumes the "L" or large sized earthquake (Mw 9.0) for the 1700
event [Wei, 2017] shows the resulting tsunami would have inundated the South Beach Douglas-fir stand.  The growth
reduction in Douglas-fir at South Beach, are less pronounced than those observed following the Japan 2011 tsunami
or the post-1700 growth suppressions observed by Jacoby et al (1997) along the Columbia River ~220 km to the north
of our study site.  While modest, spanning no more than two years relative to the control sites, and not  absolute
conclusive evidence, the growth reduction is at least consistent with such an event and with both the magnitude and
multi-year time period of growth reductions observed by Jacoby et al (1997) along the Columbia River ~220 km to the
north.

Speculatively, the observed growth reductions could possibly represent multiple large earthquakes (magnitude 8+)
over that time period, rather than one great magnitude 9 earthquake at 1700.  Although we can't rule out multiple
earthquakes using our tree ring growth data, the record of the Orphan tsunami in Japan is evidence that only one large
earthquake occurred.  Moreover, the tsunami inundation model presented here would not necessarily need to change
in the multiple earthquake scenario, since zero inundation at the old growth stand in South Beach is one possible model
result.

Although tree rings and various geologic lines of evidence have been useful in establishing the date of the last Cascadia earthquake, there is still some question as to the degree of peak ground motion associated with the event. As previously discussed, a coastal-wide inventory of liquefaction features associated with the 1700 CSZ event indicate only moderate levels of ground shaking in coastal Oregon and Washington [Obermeier and Dickenson, 2000]. Multiple, smaller magnitude megathrust earthquakes before, during, and after 1700 would be one possible explanation for a lower than expected peak ground motion, and fewer observed liquefaction features throughout the region. Mapping seismically induced landslides in the Oregon coast range is potentially another means to assess levels and distribution of seismic shaking impacts from the 1700 CSZ event, given large-magnitude earthquakes in mountainous regions around the world typically trigger thousands of landslides, and slope failures constitute a significant proportion of the damage associated with these events [Stuble et al. 2020]. Recent studies have demonstrated the utility of dendrochronology to date the ages of landslides in these settings [Stuble et al., 2020]. However, despite the Oregon Coast Range exhibiting thousands of landslides, none have been conclusively associated with the 1700 subduction earthquake, and despite proximity to the megathrust rupture, most deep-seated landslides in the Oregon Coast Range were triggered by rainfall [Perkins et al., 2019; LaHusan et al., 2020; Struble et al., 2020]. Thus a continued search for physical evidence of tsunami inundation, earthquake shaking, and co-seismic landslides is needed to refine expectations of the inundation as well as intensity and distribution of ground shaking during future Cascadia megathrust earthquakes.

**7.0 Conclusion**

We presented a series of tree-ring data from an old-growth Douglas-fir forest in South Beach, Oregon that shows significant growth reduction at the time of the A.D. 1700 Cascadia subduction zone earthquake relative to control sites. In addition, we presented a new, high resolution model of the 1700 tsunami inundation at South Beach old-growth site. Due to significant variation in the South Beach topography, several areas are predicted to experience water levels up to 17 m and a range of inundation depths up to 16 m, however the location of the old-growth stand may be subjected to a range of inundation depths from 0-10 m. To better detect tree growth anomalies near AD 1700, we also compared the South Beach Douglas-fir tree-ring data to two other Douglas-fir data sets from the Oregon Coast Range and western Cascade Mountains, which would have experienced similar climate conditions but not tsunami inundation. When compared to these control sites, South Beach tree growth is significantly lower in 1700, and reaffirms that the South Beach Douglas-fir growth is unusually low for the region. Thus the timing of the observed growth reductions in the South Beach Douglas-fir stand is consistent with these disturbances being associated with the A.D. 1700 Cascadia megathrust earthquake and the resulting tsunami, subsidence and ground motion. Overall, we think our study further supports the view that tree-ring data is a promising tool for providing insights on the spatial distribution of co-seismic impacts from megathrust earthquakes, as well as potential ground-truth information for tsunami inundation models.

## 8. Sample Availability

Upon acceptance of the manuscript, the Mike Miller, Cape Perpetua, and Marys Peak tree-ring data used in this study will be added to the NOAA National Centers for Environmental Information International Tree-Ring Databank, https://www.ncdc.noaa.gov/data-access/paleoclimatology-data/datasets/tree-ring

## 9. Code Availability

The tsunami model can be made available with a request to oar.pmel.tsunami-webmaster@noaa.gov followed by a model software training course provided by the NOAA Center for Tsunami Research

## 10. Author Contribution

RD prepared the manuscript with contributions from all co-authors. BB and RD collected the tree-ring samples, performed growth disturbance analysis, and wrote the manuscript. YW developed the tsunami model code and performed the simulations, and wrote tsunami section of manuscript. SM drafted initial maps used in manuscript.

## 11. Competing Interests

The authors declare that they have no conflict of interest.

## 12. Acknowledgements

The authors wish to thank the editor and three reviewers. We also thank J. Haxel, M. Fowler, and N. Simao for helping collect tree cores. The research in this paper was sponsored by the NOAA/Pacific Marine Environmental Laboratory, PMEL paper contribution number 5184. Yong Wei's work is funded by the Joint Institute for the Study of the Atmosphere and Ocean (JISAO) under NOAA Cooperative Agreement NA15OAR4320063, Contribution No. 2020-1084. All data is available from the authors upon request, without undue reservation, to any qualified researcher.

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

500

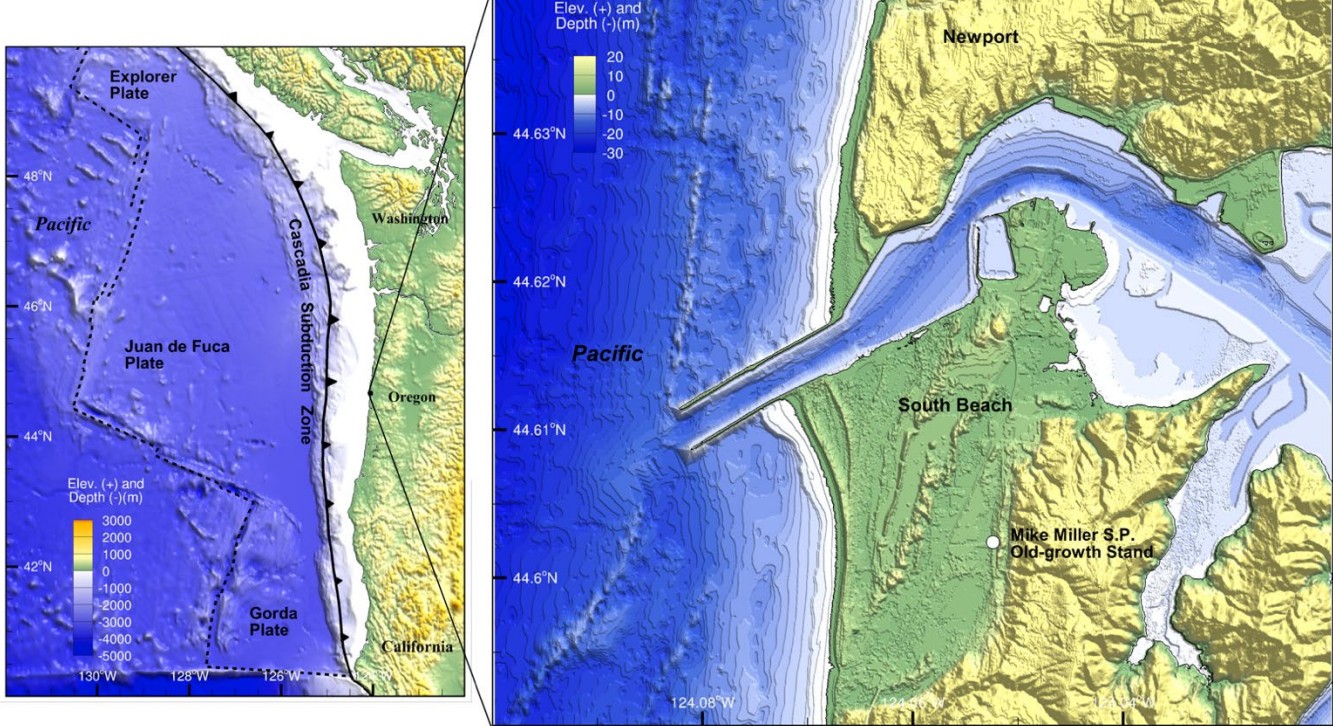

501

**Figure 1**: Map showing location of Newport and South Beach along the central Oregon coast.  White dot (at an elevation of 4m above MHHW) shows position of Mike Miller State Park in South Beach, which is location of Douglas-fir tree (*Pseudotsuga menziesii*) old growth stand whose ages extend back past AD 1700. The state park is located ~ 2 km south of the Newport-Yaquina Bay, ~1.2 km east of the shoreline and ~600 m east of Highway 101. Maps were created using digital elevation data points complied by National Center for Environmental Information (http://ncei.noaa.gov), and State of Oregon's Department of Geology and Mineral Industries (https://www.oregongeology.org/lidar/).

508

509

510

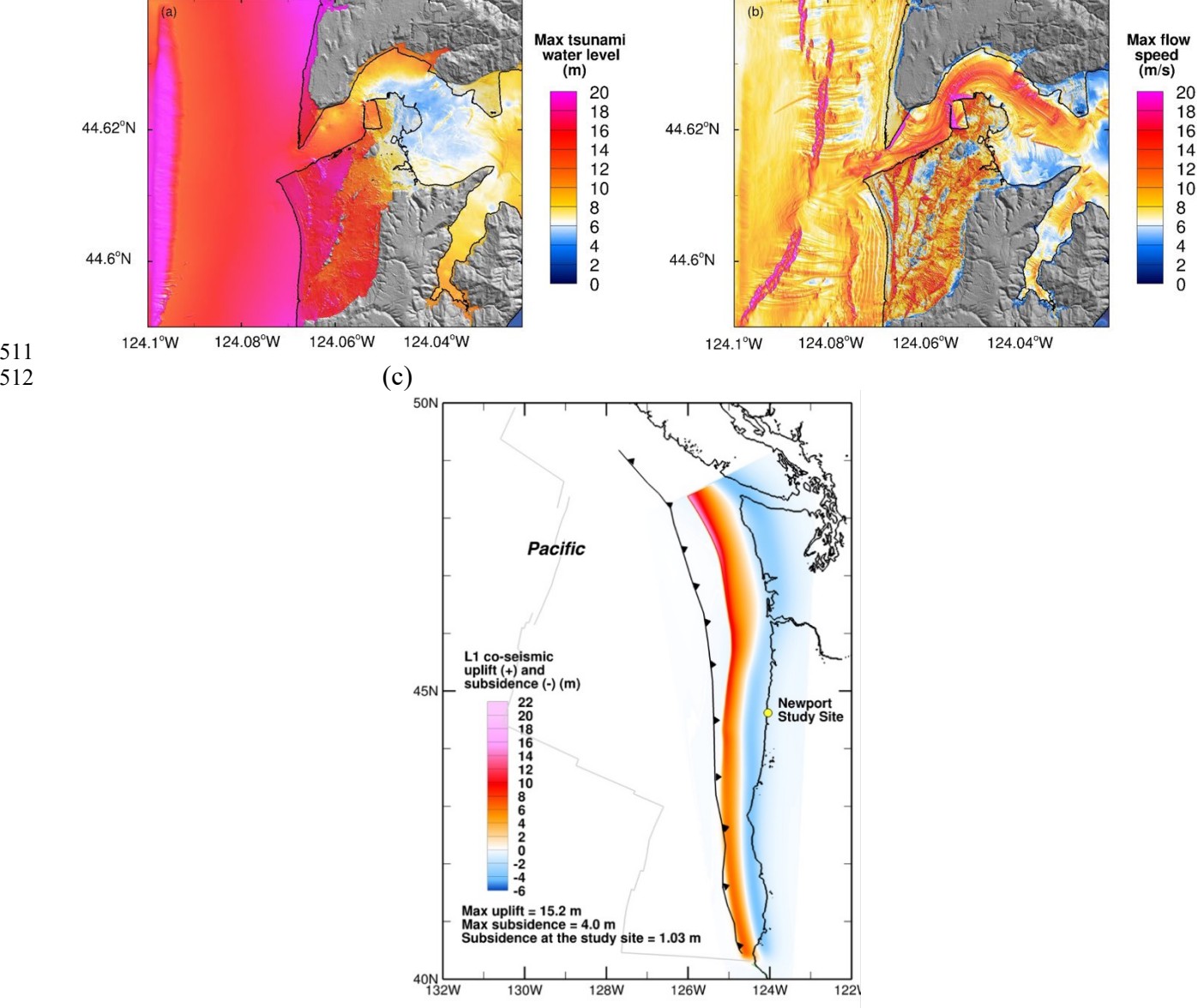

511
512

**Figure 2**: (a,b) Model of the maximum tsunami inundation level and the maximum flow speed for South Beach assuming the
"L" or large sized earthquake (Mw 9.0) for the A.D. 1700 event [Wei, 2017]. The L model assumes a finite-area fault-source
with maximum coseismic displacement of 15.2 m and subsidence of ~1.03 m at South Beach Contour levels are shown. (c)
The L model assumes a finite-area fault-source with maximum coseismic displacement of 15.2 m and subsidence of ~1.03 m
at South Beach [Witter et al., 2011].



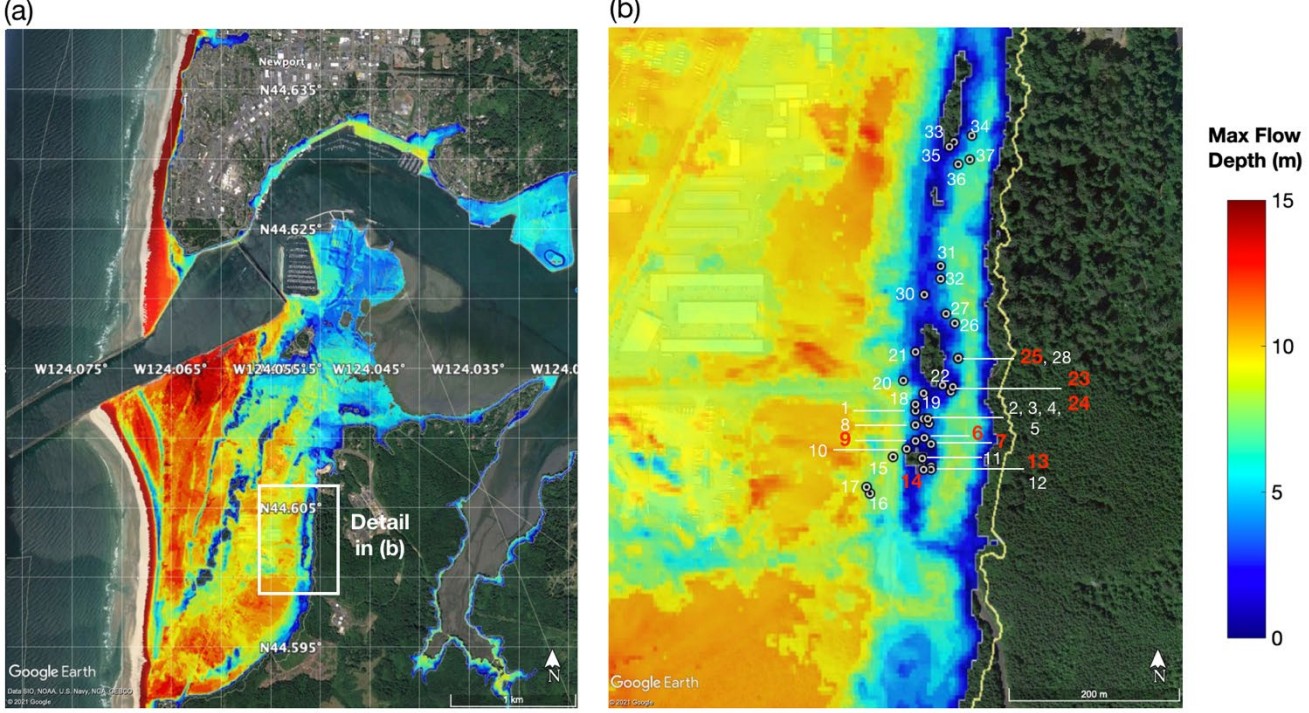

**Figure 3**: (a) Model of maximum tsunami inundation depth at South Beach for the "L" sized earthquake A.D. 1700 event; (b)
Zoom in view of the tsunami inundation depth at Mike Miller State Park. Gray dotted circles show location of trees used in
this study on north side of the Stand. Red numbers are the tree locations whose growth chronologies are shown in **Figure 4a,b**.
White numbers shows trees that were cored, but chronologies do not include the years before AD 1700. Colors on map show
inundation depth from the model, implying 0-10 m of inundation depth at the Mike Miller Park Douglas-fir stand. Green areas
are high ground locations that show no inundation. Yellow line shows, for comparison, the model of maximum run-up height
for the Mw 9.2 "XXLarge" earthquake scenario [Priest et al., 2013]. Base maps made using © Google Earth 2016.

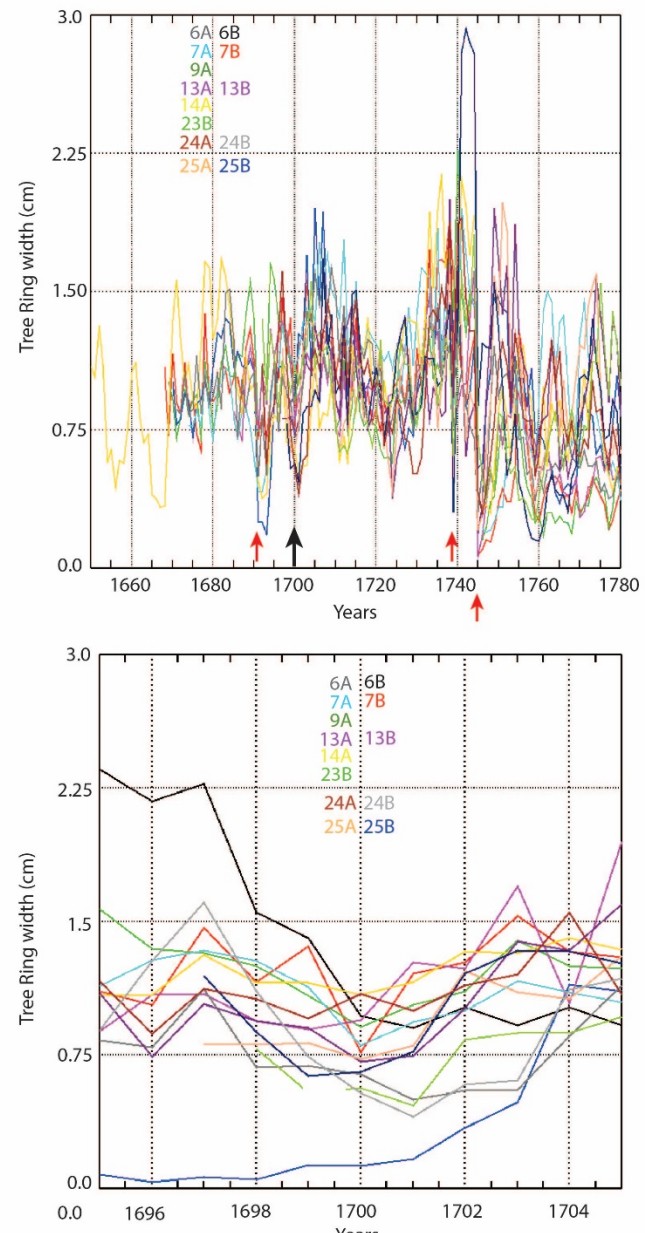



**Figure 4**: a) Tree-ring growth records of old-growth Douglas-fir trees (*Pseudotsuga menziesii*) located in Mike Miller State
Park, South Beach, Oregon (see **Figure 1**). Vertical axis shows ring growth in cm, time range covers several decades before
and after AD 1700. The color of each growth record was relates to alpha-numeric labels of individual trees shown in legend,
with location of trees shown in **Figure 3b**. Designation "A/B" represents two cores from same tree. Black arrow marks AD
1700 date, red arrows highlight the AD 1691, 1738 and 1745 large growth reductions that may have been caused by a
significant climatological events. (b) Shows detailed growth record of trees in (a) 4 years before and after 1700.

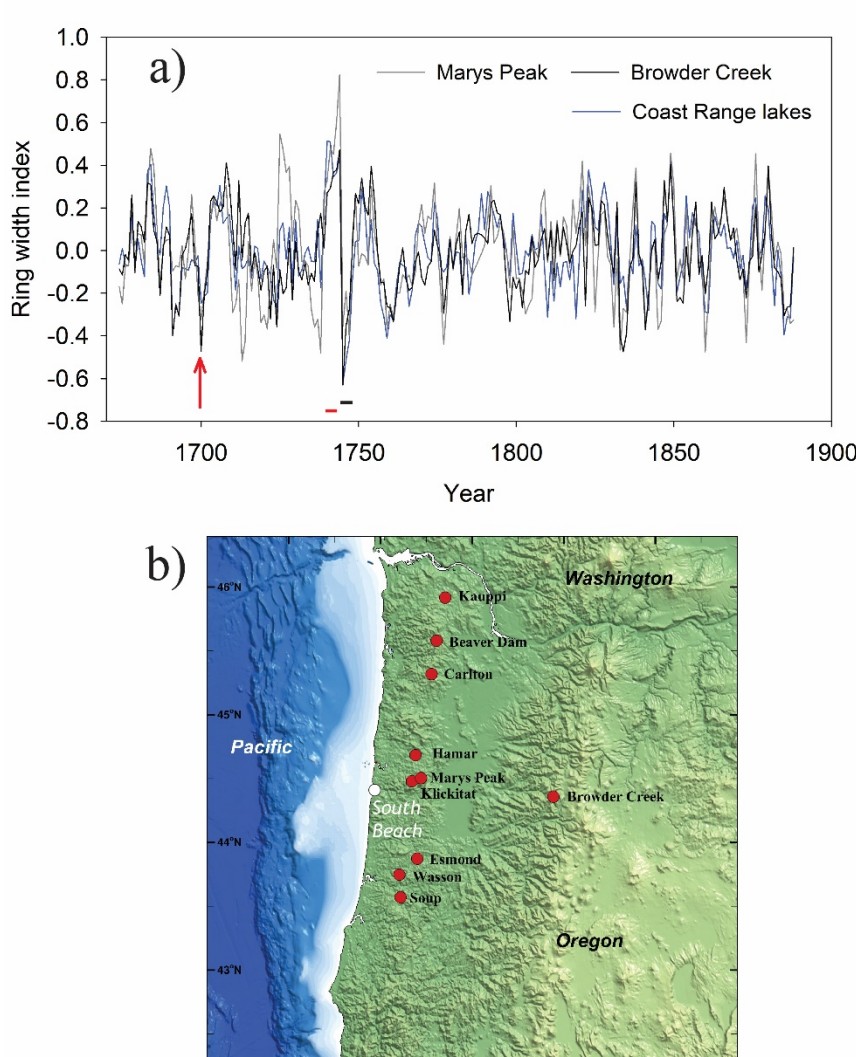

**Figure 5**. a) Difference in growth chronologies between Mike Miller and reference sites at Marys Peak, Browder Creek, and Oregon Coast Range lake sites. The A.D. 1700 chronology indicated by red arrow, the significant growth differences between coast and inland sites in 1739-1741 and 1745-1748 are highlighted by red and black lines, respectively. b) Shows location map with Marys Peak and lake sites relative to South Beach. The number of cores available at each site during the 1700 time period are South Beach (14), Marys Peak (28), Coast Range Lakes (31) and Browder Creek (30). The Oregon Coast Range lakes include: Beaver Dam Lake, Carlton Lake, Esmond Lake, Hamar Lake, Kauppi Lake, Klickitat Lake, Soup Lake, and Wasson Lake. Map created using digital elevation data points complied by National Center for Environmental Information (http://ncei.noaa.gov)
551