# Peer review of "Assessing local impacts of the A.D. 1700 Cascadia earthquake and tsunami using tree ring growth"

_Natural Hazards and Earth System Sciences, 2020_

## Author Comment (AC6)

*This is our response (texts in blue) to RC3 on "Concerning the tsunami modelling results, the figures exhibit clearly a modelling problem due to boundary effect. It should absolutely be fixed to be sure that this problem does not have consequences on both the inundation extent from the shoreline and the flow speed. It would be interesting to see the results of maximum water level and flow speed on the other nested grids to proof there is a correct junction between them. Also, it would be a good thing to add the version of MOST that has been used, with the related references (there are more recent ones available than the 1997 paper from Titov and Gonzalez). Also, as further indicated, the friction choice to set land and sea with the same value must be explained with references"*

We agree with the reviewer #3's comments regarding "the modeling problem due to boundary effect" as we also noticed some clustering of max wave amplitude close to the west boundary of the level-4 domain. To address this issue, we have carefully inspected the modeling results at level 3 and level 4, and identified the causes of the boundary effect. This seemly "unnatural" clustering of high wave amplitudes actually is due to the formation of a few short-period (periods < 1 min) waves with high amplitudes between 38-40 minutes after the earthquake origin time. These high-frequency and high-amplitude waves are often referred as "wave fissions", which can be much better simulated by Boussinesq models than by integrated shallow water equations.

As we indicated in the manuscript, our tsunami modeling is performed based on 4-level telescoped model grids. The MOST model is used only in level-1 grid and provides boundary conditions for the Boussinesq model computations in level-2, -3, and -4 grids. The boundary conditions at an inner grid level are provided along the inner-grid boundaries at grid points coinciding with their outer-grid nodes. The feeding of boundary conditions from an outer grid is done through linear interpolation in time and space of the model results in the outer grid. For example, the grid points along the boundaries of level-4 grid are provided with boundary conditions from the level-3 grid. Previously, we feed the leve-4 grid with boundary conditions at every 15 seconds. After we inspected the model results in level-3 and identified those short waves (wave fissions), we immediately realized that boundary conditions at every 15 seconds are probably too coarse for level-4 grid to fully resolve those short waves in terms of the temporal interpolation. To solve this issue, we reduced the feeding frequency of the level-3 boundary conditions into level-4 grid from every 15 seconds to every 3 seconds. This modification has greatly improved the level-4 model results and totally eliminated those "clustering" patterns close to the west boundary.

Fig. 1a shows the max tsunami amplitude in level-3 grid, where the black box indicates the domain coverage of level-4 grid. One can clearly observe large wave amplitudes (purple color) along the west boundary of level-4 grid that are produced by large short-period waves between 38 and 40 minutes after the earthquake origin time as shown in Fig. 1b.

[Figure]

Figure 1. (a) Maximum tsunami amplitude computed in level-3 grid; (b) A snapshot of the water level at 36 min after the earthquake origin time when a series of short-period waves are arriving at the west boundary of the level-4 grid.

To follow the reviewer's suggestions on seeing the results on the other nested grid to proof there is a correct junction between them, here in Fig. 2 we compare the level-3 and level-4 modeled time series at five points along the west boundary of level 4 (white dots in Fig. 1). In Fig. 2, one can see 2 to 3 short-period waves with high amplitudes between 10 and 15 m reaching the west boundary of level-4 grid between 38 and 40 minutes. The time series at all five points are nearly identical between level-3 and level-4 computations, particularly before the short-period waves appear. The level-4 computed time series show slightly larger amplitudes for those short-period waves, and this could be attributed to higher grid resolution (1/6 arc sec, or ~5 m) in level 4.

[Figure]

Figure 2. Comparison of time series of tsunami wave amplitude between level-3 and level-4 computations at points along the west boundary of level-4 grid.

Figure 3 shows the updated (a) the max tsunami wave amplitude and (b) the max flow speed computed in level-4 grid. One can see that the updated modeling totally eliminated the clustering pattern of the wave amplitude close to the boundary, and Fig. 3(a) shows the high-wave-amplitude zone along the level-4 boundary that is highly consistent with the level-3 results shown in Fig. 1. We can also see that the max tsunami amplitude and max flow speed remain

similar to our previous results in most of the areas in level 4, except the new results show higher wave amplitudes inside the bay, particularly in the inner bay.

[Figure]

Figure 3. Updated model of tsunami (a) inundation level and (b) flow speed for South Beach assuming the "L" or large sized earthquake (Mw 9.0) for the A.D. 1700 event.

The present MOST code (version 3752) utilizes the Graphics Processing Unit (GPU) technology that has led to significant reduction of computational time.

Regarding the friction factor, it is common to use a Manning's coefficient between 0.025 and 0.033 for coastal and riverine areas, and 0.03-0.04 for land surface (Chow, 1959). Presently, both MOST and the Boussinesq model of Zhou et al. are only allowed to use one constant Manning's coefficient throughout the model computation. We therefore chose an average Manning's coefficient 0.03 in our models. Our modeling experience over the years have shown that 0.03 is a reasonable approximation of the ocean bottom in deep water and the land surface. It is used in MOST-based tsunami forecast system (Titov et al., 2009; Tang et al. 2013; Wei et al, 2007 and 2013; Zhou et al., 2014). It's worth pointing out that, to be conservative, the Oregon tsunami inundation maps were developed based on a tsunami model without considering the surface roughness at all.

**Detailed comments:**
3.0 Model of AD 1700 tsunami

- *98: provide the coseismic subsidence value from Satake et al.*

  It looks like the Satake et al. (2003) subduction at South Beach is between 0.5 and 1 m (and very close to the 1-m contour) for their Splay fault 9.0 scenario, approximated from Figure 8b in Satake et al. (2003). The Splay Fault L1 scenario produces slightly larger subsidence at South Beach. We've added Satake et al. (2003)'s estimated to the text.

- *104: prefer "nested" or "imbricated" to "telescoped"*

  We've changed the wording to "one-way nested" instead of "telescoped"

- *105: "The tsunami simulation model MOST (Method of Splitting Tsunami; Titov...) used in this study is based ..."*

  We agree and explained the MOST acronym here.

- *106: "wave generation and propagation".*

  We added this to the text.

- *108: "wave dispersion"*

  We added "wave" to the text and removed "frequency".

- *109-110: "the digital elevation model (DEM)" ... (last grid level)*

  We add this to text

- *111: the spatial resolution is already indicated L.105*

  We removed repeated spatial resolution.

- *113-116: not really clear – try to make it simple or add a scheme*

  We tried to simplify text.

- *115: "above the actual MSL"*

  We added this to text.

- *124: why is the Manning's coefficient chosen identical for sea and land as it should be different. Also provide reference for the 0.03 value.*

  This is explained in our earlier response to the specific comments on modeling. References are also added.

- *128: is that possible to present a ancient map or drawing of the coast showing the lack of jetties or a document justifying your choice to remove them?*

  We are not aware of detailed ancient maps of the Oregon coast that might be useful in this context.

- *134: the elevation reached by sea water is commonly called "run-up height" and not "tsunami water level"*

  We made these corrections

- 142: "than in most"

- *144-145: please refer to the articles dealing with the impact of current on trees, especially in Japan during the 2011 Tohoku tsunami*

We added the references on tsunami current here

- *147-153: you discuss about the splay fault but do not indicate if they are considered or not in your modelling finally; this is not clear.*

The splay fault is already included in the L1 source scenario (Witter et al., 2011).

---

## Author Response (AR1)

Response to reviewers 1,2 and 3.

Reviewer 1: All concerns responded to online.

Reviewer 2

As written, this paper seems to support a full rip M9 rather than multiple events (e.g. Melgar 2021). It would be great in the discussion to discuss the potential for multiple events, or utilize this site to evaluate the potential for and event prior to or after 1700. Especially as growth suppression is seen prior to 1700, could one argue that this site is evidence for an earlier or later event?

We agree our study cannot rule out multiple earthquakes, so we revised the discussion section to read:

The tsunami inundation model presented here, which assumes the "L" or large sized earthquake (Mw 9.0) for the 1700 event [Wei, 2017] shows the resulting tsunami would have inundated the South Beach Douglas-fir stand.  The growth reduction in Douglas-fir at South Beach, are less pronounced than those observed following the Japan 2011 tsunami or the post-1700 growth suppressions observed by Jacoby et al (1997) along the Columbia River ~220 km to the north of our study site.  While modest, spanning no more than two years relative to the control sites, and not  absolute conclusive evidence, the growth reduction is at least consistent with such an event and with both the magnitude and multi-year time period of growth reductions observed by Jacoby et al (1997) along the Columbia River ~220 km to the north.

Speculatively, the observed growth reductions could possibly represent multiple large earthquakes (magnitude 8+) over that time period, rather than one great magnitude 9 earthquake at 1700.  Although we can't rule out multiple earthquakes using our tree ring growth data, the record of the Orphan tsunami in Japan is evidence that only one large earthquake occurred. Moreover, the tsunami inundation model presented here would not necessarily need to change in the multiple earthquake scenario, since zero inundation at the old growth stand in South Beach is one possible model result.

Small  comments have been addressed:

Line 9: Although the study does discuss some of the spatial components of disturbance history, it doesn't read as the principal emphasis/analysis of the tree-ring work.  I would suggest changing to "We present an investigation of the disturbance history…"

We agree, and revised the abstract.

Line 10: I would suggest "changes" rather than effects? Agree, we added "changes"

Line 22: erroneous floating period?

 Removed erroneous period.

Line 29: add "growth" before suppression

Added growth

Line 30: for non-dendrochronologists what is a "growth-event"?

Changed "event" to "increases" here.

Line 32: "Here we present a spatial analysis of the disturbance…"

Removed "spatial focused"

Line 43: replace "it seems" with "it is plausible".

We edited line to say "it is plausible"

Line 43: add in here "in the form of traumatic resin ducts, ring width suppression etc." after "might be recorded in ring widths"

*Added "in the form of traumatic resin ducts and ring width suppression" to sentence*

Line 57: "Occurred **in** the Cascadia Subduction Zone" Changed to "along"

Line 63: as these dates aren't exact, maybe add an ~ in front of the dates…..We added the "~" before the dates

Line 67: change "should" to "would"…… Changed to "would"

Line 94-95: rogue paragraph space?…… Yes there is, we removed  it

Line 111: "it" refers to the digital-elevation and bathymetric grids? The inundation model? The elevation grid, and we substituted this in for "it"

Line 170: I would define reaction wood the first time it is mentioned

Line 200: define "water-logging" or possibly show an image of the traumatic resin canals and water logging as seen with a microwood anatomy image

Line 203: Where any statistical analyses done to quantitatively look at the ring growth suppression? SEA or growth release analyses?

Line 245: How were the stand-wide releases detected? Or in this case, not detected

Line 261: It would be good to list what altitude these sites are at, as up mountain climate sensitivity of trees is often stronger

Altitude of mountain sites…Mike Miller ranges from 6 to 20 m above sea level at the stand location. So although

Line 266: Only the 50-yr splines were used on the control chronology>? Not the NEGEX? It was slightly unclear to me why NEGEX was dropped and why 50 yr spline was used.

Line 269: "trees from OR lakes" or "the lake's trees"

Line 270: Again, how was the growth anomaly detected?

Figure 4 a: are these detrended ring width?

Figure 5 b. All these lakes are not mentioned in the manuscript – it is not clear why they are all labeled.

Reviewer 3:

*We agree with the reviewer that the manuscript lacks recent references about the study of tree-ring in earthquake assessment (e.g. Allen et al., 2019; Fu et al., 2020 about the 1950-Zayu-Medog earthquake, etc.) and in tsunami assessment (e.g. Buchwal et al., 2015 for Greenland tsunami of Nov. 2000; Lopez et al., 2017 for the Tohoku tsunami; Kubota et al., 2017 for the effect of saltwater on trees, etc.). However, not all references suggested by the reviewer are relevant to our study.*

The state-of-the-art should be improved to show the common practices and methodologies in the domain and explain why they chose one way to study instead of another one. We chose a similar methodology of reviewing tree ring growth disturbances at the time of the tsunami as has been employed by the previous studies listed.

*We added the following summary of the references the reviewer suggested to the text.*

*Recent studies have demonstrated the utility of using tree-ring growth chronologies for assessment of tsunami and earthquake impacts on coastal environments. As for seawater impacts from tsunamis, Wang et al (2019) performed a regional assessment of coastal western Washington forests and demonstrated that seawater exposure drives reductions in growth, decreased carbon isotope discrimination of needle-leaf trees, increased mortality and greater climate sensitivity, regardless of whether the seawater exposure is recent or long-term. Kubota et al. 2017 studied trees growing on coastal sand dunes in Japan that were immersed by the*

*tsunami that following the great Japan Earthquake on 11 March 2011. Trees that survived direct physical damage from the tsunami, began to die the following summer, likely due to the physiological stress of salt water immersion. Tree rings that were immersed in seawater from the tsunami had higher $\delta^{13}C$ values in the earlywood that formed in the spring following the tsunami than those formed prior to the disaster.*

*As for tree growth disturbances due to earthquake shaking, Fu et al. (2020) showed how the 1950 Zayu-Medog magnitude 8.6 earthquake, which devastated the southeastern Tibetan Plateau, influenced tree growth, where 60% of sampled trees showed growth responses during period 1950–1955. However, alpine trees near the treeline were less disturbed than those located at mid and low elevations. Severe growth suppressions occurred during the first three years after the earthquake, and were stronger at low elevations. Growth rates returned to pre-earthquake values 45 years after the earthquake occurrence. Buchyal 2015 also used dendrochronological methods for dating and assessing the environmental impacts of tsunamis in polar regions following the 21 November 2000 landslide-generated tsunami in west Greenland. Samples of the shrub Salix glauca (greyleaf willow) reveal unambiguously that the tsunami-impacted area was immediately colonized during the following summer by rapidly growing shrubs. Moreover, control site specimens recorded damage at the time of the tsunami, which was likely due to high seasonal -rainfall that contributed to the tsunami-generating landslide.*

Detailed comments:

Abstract

- Douglas with only one "s" -*We removed the extra "s"*
- 9: add latin name Pseudotsuga menziesii – *Added latin name*
- 10: add "(CSZ)" after Cascadia subduction zone – *Added (CSZ)*
- 13: "0 to 10 m" *Edited*
- 14: "shows that several trees experienced" *Added "that"*
- 19: why do you indicate 110 years and not 320 or more ? it should be clarified for the reader even in the abstract – maybe better to indicate between 1660 and 1780, referring to the period you analyzed

  *Added to the sentence >321 year growth history…*
- 19: remove "." after "location" *Removed*

Introduction

- 24: "along the Sumatra and Japan coasts" *Changed to Japan*

- 26-29: there is plenty of recent interesting papers, especially from Japanese teams, dealing with those subjects; you must add references here. *We added these references to the Introduction section.*

- 29: not sure that the term "suppression" means what you want to write. Please review this carefully. *We reviewed this term, suppression means hindered growth in the context.*

- 36: "Cascadia Subduction Zone" – add (CSZ) and use it in the rest of the paper. *We added (CSZ).*

- 38: Sometimes you talk about the 1700 Cascadia Subduction Zone earthquake, sometimes to the 1700 megathrust earthquake, etc. Please standardize. *We interchange these words to avoid repetitive text, but did edit to try to standardize.*

- 40: "in the coastal range" *Added coastal range*

- 43: replace "the ring widths of trees" by "the width of the tree rings" (and use the same wording everywhere). *We revised this sentence.*

- 45-47: and elsewhere? There are papers and technical reports available focusing especially on tree-ring analysis in earthquake research in other parts of the World that could help your demonstration (Arsdal et al., 1998; Wells and Yetton, 2004; Stoffel and Bollschweiler, 2008 in the same journal : https://nhess.copernicus.org/articles/8/187/2008/, etc.). Please refer to some of them to show at least a summary of the state of the art.

  *The point of the summary here is to state there is little tree ring work done along the Oregon coast. The paper suggested here are useful, and we added these to the first part of the Introduction where we describe global tree ring studies.*

48: remove space after "." *Removed.*

- 50: you indicate that the tsunami may cause physical damage to trees but what about the chemical damage? Probably a way to explore in Yoshii et al. (2012; https://link.springer.com/article/10.1007/s00024-012-0530-4)

  *We are aware of the chemical impact on trees from exposure to seawater. We note this in the text, but feel any more detail on this subject is outside the scope of our study. Also the Yoshii et al 2012 study was very interesting, the ion discriminant method for detection of tsunami inundation is best for areas of limited rainfall, which is not the case for the Oregon coast range.*

  52: "and where there is" *Where good inundation models exist.*

- 53: where are these "large population and municipal infrastructure" ? please locate on one of your figures and refer to it in the text. *This would be the town of Newport*

*and South beach Oregon, with as shown in Figure 1. We did add >10,000 people to specific our definition of large.*

*2.0 Evidence for megathrust earthquakes and tsunamis:*

- 57: "On January 26, 1700" or "On the 26th of January, 1700" and remove "in the year 1700 AD" *We removed this text.*

- 58: either write "plate boundary" or replace with "plate interface" *We replaced this*

- 61: replace "The 1700 earthquake" by "It" (apply this in other parts of the document) *We replaced with "It"*

- 63: please add the map locating approximately the epicenter of the earthquakes. *This is not known*

- 64: "comprise" – strange word, please change it. *Comprise means "consist of; be made up of". We changed to "makes up".*

- 65: Simplify your sentence, for example : "The 1700 Cascadia earthquake ground motion and … are modelled from ~05 to 1.2 g … . The shaking during this event should … ". *We changed these sentences as suggested.*

- 69-70: This sentence is a bit strangely located. You should detail which timing you're looking for. If it is the date, what I expect, please indicate why. *We are discussing earthquake origin time We changed the beginning of the sentence.*

- 71: "the dates have been obtained from". *We rephrased this sentence.*

- 73-75: it would be interesting to have a map of those coastal forests – maybe add their location on one of your figures. *We agree, it is interesting, however the location will not be on our existing location maps, and would require an additional figure. We do, however, note the location relative to our study within the text.*

- 81 and after: you discuss about the liquefaction but you should above all highlight that the main question to which this study tries to answer is: what has been the impact of the 1700 earthquake? And for this, different methodologies have been applied, like looking for liquefaction features, and looking at the tree-ring growth.

- *Added the following sentence to the beginning of the paragraph at line 81: Additional methodologies have been employed to assess the coast-wide impacts of the 1700 earthquake. For example, a coastal-wide inventory*

-

*3.0 Model of AD 1700 tsunami*

- 98: provide the coseismic subsidence value from Satake et al. *We added the Satake estimate (19 m) here.*

- 104: prefer "nested" or "imbricated" to "telescoped" *We disagree, splay is a structural geology term for this type of subduction zone faults, and would prefer to keep this term here.*

- 105: "The tsunami simulation model MOST (Method of Splitting Tsunami; Titov…) used in this study is based …" *We agree and explained the MOST acronym here.*

- 106: "wave generation and propagation". *We added this to the text*

- 108: "wave dispersion" *We added " wave" to the text.*

- 09-110: "the digital elevation model (DEM)" … (last grid level). *We add this to text*

- 111: the spatial resolution is already indicated L.105 *We removed repeated spatial resolution.*

- 113-116: not really clear – try to make it simple or add a scheme. *We tried to simplify text.*

- 115: "above the actual MSL". *We added this to text.*

- 124: why is the Manning's coefficient chosen identical for sea and land as it should be different. Also provide reference for the 0.03 value.

- 128: is that possible to present a ancient map or drawing of the coast showing the lack of jetties or a document justifying your choice to remove them? *We are not aware of detailed ancient maps of the Oregon coast that might be useful in this context.*

- 134: the elevation reached by sea water is commonly called "run-up height" and not "tsunami water level"

- 142: "than in most"

- 144-145: please refer to the articles dealing with the impact of current on trees, especially in Japan during the 2011 Tohoku tsunami

- 147-153: you discuss about the splay fault but do not indicate if they are considered or not in your modelling finally; this is not clear.

*4.0 Impacts of Earthquakes…:*

- 160: add references.

- 169: remove space after "." *Removed*

- 171: "Fort Tejon" *spelled out "Fort"*

- 179: add the latin name Picea sitchensis – end of sentence not clear, please rewrite. *Not sure what this means…we tried to rephrase sentence to be clearer.*

- 201: which reaction? Please develop. *We add growth to the sentence for clarification.*

- 205-208: what about the effect of salt in the soil and thus in the tree growth? Several studies available to deal with this problem. *We added a sentence and a few references here to address impacts from seawater.*

*5.0 Tree ring growth…:*

- 229: detail what is COFECHA on one sentence to show that is adapted for such verification. *We added: Crossdating was then statistically verified using the program COFECHA, which is designed to find errors in chronologies [Holmes 1983].*

- 233: idem for ARSTAN. *We added.. a tree-ring standardization program based on detrending and autoregressive time series modelling*

- On L.226 you indicate that two cores were collected from each tree but only 12 from 8 trees at L.235… *please clarify. We meant, 1 to 2 cores were collected from each tree. We revised text to clarify.*

- 250: show the 5 growth reductions on the figure (only 4 arrows). The other disturbances will not be in the time frame of the figure. *We changed this to read "several other disturbances..but the most notable are at* 1691 and again in 1739 and 1745

- 251: (arrows on Figure 4a).. *We added (the arrows on figure 4a)*

- 252-253: you must show a comparison between the two dataset – maybe adding the curves on the same figure / two separate figures are not easy to compare. *We appreciate the reviewers comment, but to add figures 4a and 4b to the same plot would make this a very busy plot. The long term record (4a) can be used to see disturbances at other dates* 1691, 1738 and 1745. Figure 4b highlights the largest growth reductions at 1700.

  268-271: idem – show figure with comparison. *I assume this means show figure 5 without* the detrended by 50-year splines.

-

-

*6.0 Discussion:*

- 276: "to other inland sites" *We deleted the ","*
- 278: same remark about "suppression" – please change word  *-Change to reductions*
- 294: "another mean to" – *We changed to "another means to assess"*
- 291-294: check and refer to Perkins et al. (EOS, 2018) – *We checked and added the reference*

*Summary:*

- Replace "summary" by "conclusion" *Replaced*

- 316: it would be great to add a final sentence like this one: Coastal trees, especially old ones, should be preserved from logging to help to reconstruct the seismological and tsunamical history of a region, as well as they provide natural coastal protection.

- *We like this statement, but it is somewhat of a political naturel, and we'd prefer not to add it*

- 320: "in this study will be added in ..." *We added "added" to the text*

*References:*

The list of references must be standardized referring to NHESS guidelines. Also, the DOI linked should be added when it is possible (this is the case for most of the references). *Added DOI when possible*

*Figures:*

Figure 1:

- Add a small map located Oregon, at least in the US.

  *Oregon is labelled on the map of the west coast od the U.S. and in our opinion a small map is not needed.*

- 427: add altitude value of the stand (mean value) - "data points compiled"

  Figure 2: *The relative "altitude" or elevation of the stand is apparent by comparison of the color water level map provided.*

- a) and b) must have different colour scales to avoid confusion

  *We disagree, the same color scale aids with relative comparison between the diagrams.*

- There is a serious boundary effect on the left edge of the grid which leads to strange high frequencies pattern on both water level and flow speed maps. You should fix the problem before publication.

  *We've identified the issue and re-run the model. The reason for those "clustered" high amplitudes is due to a less-frequent (previously at every 15 sec) feed of modeled boundary conditions from the outer grid. Inputting boundary conditions at every 15 sec is enough in most of the model situations. However, this is a special case that a few very short-period (<1 min) dispersive waves (we also call the phenomenon "wave fission") developed offshore ~30 min after the source rupture, and it needs more frequent input from the outer layer to fully capture those short waves. The new run has totally eliminated those wave clusters, and*

*shows a continuous high wave zone near the boundary. We revised the text and figures to show the new model*

- Also, it should be indicated whether it shows representation of the maximum water level reached on each point of the grid upon the simulation time, as well as the maximum flow speed, or if it corresponds to values at a time = i. *The Values pertain to the maximum values in the grid simulation, and we noted this in the text.*

- Coordinates should be out of the maps to clarify. *We disagree and like the appearance of the coordinates within the maps.*

  Figure 4a and 5a: the y axis should be the same on both figures to help the reader to compare easily. But my previous comment was to show the two on only one figure.

  *We made figures 4a and 4b to have the same vertical axis. Figure 5 is a separate plot form figure 4, represents a normalized scale, and we'd like to keep the axis as is.*

**Citation**: https://doi.org/10.5194/nhess-2020-427-RC3